# Effect of BASC and BASCA Heat Treatment on Microstructure and Mechanical Properties of TC10 Titanium Alloy

**DOI:** 10.3390/ma15228249

**Published:** 2022-11-21

**Authors:** Mingyu Zhang, Xinbing Yun, Hongwang Fu

**Affiliations:** Engineering Research Center of Continuous Extrusion, Ministry of Education, Dalian Jiaotong University, Dalian 116028, China

**Keywords:** BASC heat treatment, BASCA heat treatment, microstructure, tensile property

## Abstract

The purpose of this study is to investigate two new heat treatment processes on the mechanical properties of TC10 titanium alloy. By changing the β annealing temperature, the variation in microstructure and mechanical properties of TC10 titanium alloy were investigated. The results showed that with the increase in β annealing temperature the microstructure type changes from an equiaxed structure to a lamellar structure. The strength of the alloy then increases firstly, followed by a decrease, while the plasticity decreases all the time. Microstructure observation revealed that the alloy is uniformly composed of α phase and β phase after the two processes. In addition, it was found that the fracture morphology of the equiaxed structure is mainly dimples, showing ductile fracture characteristics, while the fracture morphology of lamellar microstructure is mainly crystalline, showing brittle fracture characteristics. These results indicated that reasonable β annealing temperature can be set according to different requirements to obtain different types of microstructure and mechanical properties, which expands the application field of TC10 titanium alloy.

## 1. Introduction

Titanium and titanium alloys exhibit many excellent properties such as high specific strength, good biocompatibility, non-magnetic properties, and good corrosion resistance [1,2], and thus are widely applied in military industries, marine engineering, aerospace, biomedical and other fields [3,4]. TC10 titanium alloy (Ti-6Al-6V-2Sn or Ti662) possesses higher strength alloy, which was developed based on TC4 titanium alloy by adding more β stable elements [5]. The alloy has excellent mechanical properties, heat resistance, oxidation resistance and corrosion resistance [6]. Therefore, the alloy is widely used in many fields such as aerospace, marine engineering, petroleum engineering and so on [7]. The alloy is similar to TC4 titanium alloy. It has good thermal processing performance and can be processed into bars, plates and wires by forging, rolling and other processes [8].

The strengthening methods of titanium alloy are mainly deformation strengthening and heat treatment strengthening. Heat treatment strengthening is one of the most widely used methods because of its fast, convenient and energy-saving characteristics. At present, the heat treatment methods of TC10 titanium alloy mainly include ordinary annealing, double annealing, isothermal annealing, solution aging and so on. In recent years, although domestic and foreign scholars have carried out a lot of heat treatment strengthening research on TC10 titanium alloy [9,10,11], most of them are still based on traditional heat treatment processes such as solution aging, and other heat treatment processes are less studied.

In recent years, with the increasing application of TC10 titanium alloy, the traditional heat treatment process has difficulty matching the rapid development of TC10 titanium alloy. It is thus necessary to propose new heat treatment processes. BASCA (β annealing + slow cooling + aging) heat treatment is a new type of heat treatment process, which is commonly used in β-type titanium alloys [12]. Figure 1 shows the schematic diagram of the BASCA heat treatment process. The process can be divided into three stages: firstly, the alloy is heated to the β phase region for heat preservation (BA stage); then, it is cooled to a certain temperature with the furnace and heat preservation, and then cooled to room temperature by air cooling (SC stage); and finally, an aging treatment is performed (A stage). Although some investigations have studied BASCA heat treatment process [12,13] on β-type titanium alloy, there are few studies on how the BASCA heat treatment process affects the microstructure and tensile properties of α + β-type alloy. Therefore, in this paper, the BASCA heat treatment process is divided into two processes: BASC and BASCA, and different types of microstructures are obtained by changing the temperature of the BA stage. Then, the tensile properties are tested to study the effects of BASC (β-annealing + slow cooling) and BASCA heat treatment processes on the microstructure and mechanical properties of TC10 titanium alloy. The research results are helpful to better understand the relationship between heat treatment process, microstructure and tensile properties, thus formulating the best heat treatment process parameters and expanding the application field of TC10 titanium alloy.

## 2. Materials and Methods

TC10 titanium alloy bar with a length of 300 mm and a diameter of 103 mm was used as the test material. The raw materials of the alloy were small particle sponge titanium and intermediate alloy, which were melted into an ingot by a vacuum self-consumption melting furnace (VAR) for three times, and then the bar was forged by multi-fire forging. The chemical composition of the alloy, measured by ICP-AES analyzer, was 5.7% Al, 5.6 % V, 2.3% Sn, 0.65% Fe, 0.65% Cu, 0.121% O and Ti allowance. The phase transition temperature of the alloy, measured by continuous heating metallographic method, was 940 °C. The original microstructure of the alloy is shown in Figure 2, in which the morphology of the α phase is divided into two types: one is the primary α phase (α_p_) uniformly distributed in the matrix, and the morphology is mainly equiaxed; the other is the secondary α phase located in the β transformation structure (β_T_), and the morphology is mainly fine and needle-like. Therefore, the original microstructure of the TC10 titanium alloy is a bimodal structure formed by forging in the two-phase region, which is composed of the primary α phase and the β transformation structure. The β transformation structure is composed of the secondary α phase and the residual β phase between the secondary α phase.

TC10 titanium alloy was processed into 8 parts by sawing machine and wire cutting, and then BASC and BASCA heat treatment were carried out by box resistance furnace with precision grade 2. The specific heat treatment system was shown in Table 1. When the heat treatment was completed, the sampling process was carried out. In order to ensure the consistency and scientificity of the test, the sampling direction of the tensile sample was the longitudinal direction of the bar. Two samples were tested in each group of tensile tests, and the average value was taken as the test result. The specific sampling position and tensile sample size are shown in Figure 3 and Figure 4, respectively.

The optical microscopy (ICX41M) was used to observe the microstructure at low magnification. The SUPRA 55 field emission scanning electron microscopy was used to observe the microstructure at high magnification and the tensile fracture morphology. The Panaco Empyrean X-ray diffraction was used for XRD test, and the scanning angle was set at 20–80°. The XRD test results were analyzed by Highscore plus software (version: 3.0d). The JEM 2100F transmission electron microscopy (Japan Electronics, Tokyo, Japan) was used for high-resolution morphology observation and crystal structure analysis. The tensile test was performed by GNT100 electronic universal testing machine (NCS, Beijing, China), and the test standard was implemented according to GB/T 228.1-2020. The test data were tensile strength *R*_m_, yield strength *R*_p0.2_, elongation after fracture *A*% and reduction in area *Z*%. The extensometer was used during the test.

## 3. Test Results and Discussion

### 3.1. Microstructure

Figure 5 is the microstructure of TC10 titanium alloy after BASC heat treatment. Compared with the original microstructure, the morphology of α phase in the microstructure changes obviously. In Figure 5(a_1_,a_2_), the size of equiaxed α phase coarsens and grows up, and lamellar α phase appears. The content and size of equiaxed α phase in Figure 5a_3_ decreases obviously, while the content and size of lamellar α phase increases. In Figure 5a_4_, the equiaxed α phase has completely disappeared, and coarse β grains and relatively complete grain boundary α phase appears in the microstructure. At the same time, a large number of clusters are formed in some coarse β grains, and the lamellar α phases in the clusters grow interlaced or parallel, and spherical α phases are formed at the grain boundaries.

When the BA temperature is in the two-phase region, before the start of the SC phase, there is a part of the α phase in the microstructure, and the β→α phase transition occurs during the SC phase. At this time, there are two possibilities for the precipitated α phase. One is that the precipitated α phase nucleates and grows at the β grain boundary, and the other is that the precipitated α phase nucleates and grows at the interface of the original α phase and is connected with the original α phase [14]. The orientation of the α phase precipitated at the original α interface is different from that of the original α phase, and its size is related to the cooling rate. When the cooling rate is faster, the morphology is mainly needle like. When the cooling rate is slower, the morphology is mainly plate like. During subsequent heat preservation process, the size of the α phase will further increase and stabilize. In the process of air cooling, the morphology of α phase will be retained due to the fast cooling rate.

When the BA temperature is in the single-phase region, the α phase in the microstructure has been completely transformed into the β phase. In the SC stage, the re-precipitated α phase nucleates at the β grain boundary and grows gradually to form the grain boundary α phase. In the subsequent heat preservation process, the α phase will expand from the grain boundary position to the interior of the grain, and its morphology is dominated by sheets. The expansion process is until each lamellar α contacts with each other [15]. The group comprised a group of lamellar α phases which are parallel to each other and have the same orientation in the colony is called α cluster [16]. The α cluster usually presents discontinuous or continuous lamellar morphology, and there is residual β phase between the lamellar α phases in the α cluster. When the temperature of the BA stage is higher and the cooling rate is slower, the formed sheet α is thicker, and the α cluster size is larger. At the same time, some α phases in the microstructure will nucleate and grow independently at the grain boundary to form spherical α phases.

Figure 6 shows the XRD diffraction pattern of TC10 titanium alloy after BASC heat treatment. After heating treatment, TC10 titanium alloy mainly underwent β→α phase, β→α′ phase and β→α″ phase transitions during the cooling process. The β→α′ phase and β→α″ phase are martensitic transformations, which are transformed from one crystal structure into another crystal structure by shear, and this is a non-diffusion phase transition. The β→α phase is an atomic diffusion phase transition. As the α phase and α′ phase have an identical structure (HCP) exhibiting fine needle-like morphology (position *A*), and the lattice constants of the two phases are very close, the diffraction peaks of the two phases in the XRD diffraction pattern are very close and difficult to distinguish. In addition, the main structures of the two phases are consistent, and the position of the diffraction spots is almost completely coincident, which makes the identification process more difficult. Relevant literature pointed out that the cooling rate is an important basis for judging whether martensitic transformation occurs [17]. When the cooling rate is fast, the elements in the β phase cannot be precipitated in time, and the transformation of the β phase will be carried out in the form of shear to form the α′ phase. When the cooling rate is slow, the alloy elements in the β phase are fully diffused, and the α phase is formed. In the BASC heat treatment process the SC stage is furnace cooling with a slower cooling rate, and after the subsequent holding stage at 800 °C the cooling method is air cooling. At this time, the cooling rate is slow and the undercooling is small. Because the α′ phase is obtained by the transformation of the high-temperature β phase [18], it can be determined that the needle-like phase formed in the microstructure is α phase. Therefore, the XRD diffraction peak in Figure 6 is the α phase diffraction peak. As the *a* axis ≠ *b* axis in the crystal structure of the α″ phase, the α″ phase reflects different crystal plane spacing. In the low angle range (41~42°) and high angle range (51~54°) and (61~65°) in the XRD diffraction pattern, (021)α″, (022)α″, (200)α″, (130)α″ diffraction peaks can be utilized to determine whether there are α″ phase [17,19]. It can be seen from Figure 6 that after BASC heat treatment there is no diffraction peak of the above α″ phase in the XRD diffraction pattern, indicating that there is no α″ phase in the microstructure.

The higher the heating temperature is, the more sufficient the driving force of martensitic transformation is. Therefore, the sample with BA temperature of 960 °C was selected for TEM analysis to verify the above analysis. The α′ phase is mainly distributed on the matrix, while the α″ phase is also present in the residual β phase in addition to the distribution in the matrix. These two regions are selected for electron diffraction spot marking. The specific morphology is shown in Figure 7. It is found that the target area is mainly α phase and β phase, and there is no α′ phase and α″ phase.

Figure 8 shows the microstructure of TC10 titanium alloy after different BASCA heat treatment. Compared with BASC heat treatment, the alloy undergoes an additional aging treatment, resulting in a more stable microstructure of the alloy. The overall change trend is consistent with BASC heat treatment. After BASCA heat treatment, the staggered region of some fine needle-like α phases can be seen between the lamellar α phases. This is because during the aging process, the metastable β phase formed during BASC heat treatment will be decomposed to form smaller lamellar and needle-like α phases [20]. Compared with the BASC process at the BA temperature of 960 °C, many clusters are formed in all the original β grains, and the clusters scale is larger.

Figure 9 shows the XRD diffraction pattern of TC10 titanium alloy after BASCA heat treatment. As the alloy is cooled to 800 °C by furnace cooling (not cooled to room temperature) in the BASC heat treatment process, a small amount of decomposable residual metastable β phase is retained in the microstructure. During the aging process, the residual metastable β phase is transformed into α phase and β phase. This decomposition transformation process is allotrope transformation. Combined with Figure 8, it can be seen that the diffraction peak of the weak (200)β phase appears in the XRD diffraction pattern, which further verifies the above analysis.

Summarily, during the heating process, the content and size of the equiaxed α phase will gradually decreases with the increases in temperature. When the temperature reaches the phase transition point, the equiaxed α phase completely disappears. At the same time, the content and size of flaky α phase increases continuously. Therefore, when the BA temperature is at the α + β two-phase region, the microstructure type of the alloy changes from bimodal structure to equiaxed structure after the BASC and BASCA heat treatments. When the BA temperature is β single-phase region, the microstructure type changes from bimodal structure to lamellar structure.

### 3.2. Tensile Properties

Figure 10 shows the tensile properties of the alloy after BASC and BASCA heat treatments, and Figure 11 shows the engineering stress–strain curves. It can be found that the *R*_m_ and *R*_p0.2_ of the alloy after BASC heat treatment increase first, and then decrease with the increase in BA temperature. When the BA temperature is 940 °C, the strength of the alloy reaches the maximum with *R*_m_ being 1083 MPa and *R*_p0.2_ being 950 MPa. The *A* and *Z* of the alloy gradually decrease with the increase in BA temperature and reach the minimum value when the BA temperature is 960 °C, where *A* is 12% and *Z* is 19%. After BASCA heat treatment, the change trend of strength and plasticity of the alloy is consistent with that of BASC heat treatment. The strength reaches the maximum value when the BA temperature is 940 °C, where *R*_m_ is 1054 MPa and *R*_p0.2_ is 874 MPa, and the plasticity reaches the minimum value when the BA temperature is 960 °C, where *A* is 11.5% and *Z* is 27%.

In general, under the action of external stress, when the stress value is greater than the critical stress required for the dislocation to start, the dislocation begins to slip. In the process of slip, the different morphology α in the microstructure has different degrees of obstruction to the dislocation slip. The equiaxed α phase in the microstructure is the main factor affecting the plastic properties of the alloy. This is because the equiaxed α phase contains more movable slip systems, and the deformation coordination is good, which promotes the slip. When the alloy is plastically deformed, the slip will be activated first in the equiaxed α phase with the largest orientation factor. When the microstructure contains more equiaxed α phase, deformation can be quickly dispersed into more grains, which will avoid its accumulation in a small number of grains, and then produce stress concentration, resulting in cracks and fracture of the alloy [21]. Therefore, when the microstructure contains more equiaxed α phase, the alloy can undergo large deformation so the plasticity is better. With the increase in BA temperature, the content of equiaxed α phase in the microstructure gradually decreases, resulting in the decrease in plasticity of the alloy.

When the BA temperature is in the two-phase region, the strength of the alloy is opposite to the plastic change trend. As the plasticity decreases, the strength increases. This is because with the increase in BA temperature, the equiaxed α phase in the microstructure decreases, while the lamellar α phase increases, increasing the obstacles encountered by dislocations in the slip process. When the dislocation slips, the staggered lamellar α phase plays a greater hindering role, resulting in more difficult plastic deformation and increased strength. When the BA temperature is in the single-phase region, the strength of the alloy decreases. This is because the equiaxed α phase in the microstructure completely disappears at this time, and the morphology is mainly composed of coarse α clusters. The α clusters have a greater hindrance to slip. When the alloy is plastically deformed it is difficult for dislocations to pass through the α/β phase interface smoothly, and a large number of dislocated accumulations are generated at the interface. Stress concentration is easy to occur at the interface, which makes the deformation enter the yield stage in advance, showing a decrease in strength [22].

Comparing the two groups of processes, it can be seen that the strength of the alloy after BASC heat treatment is higher than that of BASCA, but the plasticity is lower. This is because the metastable β phase in the alloy will be decomposed into α phase and β phase after BASCA treatment. During this period, the nucleation and growth of the α phase will occur. Combined with the microstructure of Figure 5 and Figure 8, the equiaxed structure has a higher degree of equiaxation in Figure 8 and the lamellar structure contains completer and more stable α clusters, which achieves the effect of stabilizing the structure. This indicates that the BASCA process mainly focuses on stabilizing the microstructure rather than strengthening the alloy compared with the BASC process.

Compared with the traditional solution of aging heat treatment, it was found that the strength of the alloy after BASC and BASCA heat treatments is lower, but the plasticity is higher. Therefore, it can be used for aircraft structural parts with lower strength and higher plasticity, such as aircraft skin and heat insulation board. It is also found that after these two processes, the alloy can form equiaxed structure and lamellar structure that cannot be obtained by solution aging treatment. Therefore, different process parameters can be formulated according to the actual engineering requirements to obtain the required microstructure and properties.

### 3.3. Tensile Fracture Morphology

Figure 12 shows the morphology of tensile fracture. When the BA temperature is in the two-phase region (Figure 12a_1_,a_2_,b_1_,b_2_), the micromorphology of the fracture is dominated by dimples (position *B*). The dimples are because during the tensile process of the alloy, due to the fast strain rate, some dislocations in the microstructure produce stress concentration, which in turn triggers micropore nucleation. As the tension progresses, the repulsive force of the dislocation begins to decrease, and some dislocations will be pushed into the micropores, activating the dislocation source again. This leads to the continuous formation of new dislocations. The newly formed dislocations are continuously pushed into the micropores, and the micropores will gradually grow in this process. Micropores will gradually converge at the fracture position, leaving traces and eventually forming dimples [23].

In general, the number and size of the dimples can reflect the plasticity of the alloy. When the number and size of the dimples are large, the plasticity is good; when the number and size of dimples are small, the plasticity is poor. It can be found that when the BA temperature is in the two-phase region, the number of dimples in the fracture microstructure is large and deep. This is because the slower cooling rate and higher holding temperature in the SC stage will make the recovery and recrystallization in the microstructure more thorough, forming more equiaxed α phases and improving the plasticity of the alloy. Thus, the microstructure is dominated by ductile fracture. At the same time, it was found that there are a certain number of secondary cracks (position *C*) in addition to many dimples in the fracture micromorphology. Both the secondary crack and the main crack propagate along the main direction of the tension, and there are finer microcracks at the end of the secondary crack, which indicates that the conditions for crack propagation are easy to meet. When the crack propagates in the microstructure, it can propagate along the α/β grain boundary junction position and can also propagate through the grain so that the crack propagation path is relatively smooth, indicating that the plasticity of the alloy is high [24].

When the BA temperature is in the phase transition temperature (Figure 12a_3_,b_3_) there are still dimples in the fracture morphology, while the number is small, and the depth is shallow. In addition, there is a tendency of the fracture to develop into a rock-like morphology, which is consistent with the characteristics of the BA temperature at the phase transition point. When the BA temperature is in the single-phase region (Figure 12a_4_,b_4_), the fracture morphology of the two processes is mainly crystalline. The dimples are small and shallow. There are obvious dissociation steps (position *D*) and obvious tearing edges (position *E*), and a small number of smaller dimples are distributed on the surface, which is a brittle fracture morphology characteristic. This is because after the single-phase region heat treatment of the alloy, the equiaxed α phase in the microstructure completely disappears and the coarse β phase with α clusters is formed. During the deformation process the grain boundary position is easier to form voids and expand rapidly, which leads to the decrease in the plasticity of the alloy.

## 4. Conclusions

In this paper, the effects of BASC and BASCA heat treatments on the microstructure and mechanical properties of TC10 titanium alloy were studied. By changing the BA temperature, the relationships among heat treatment process, microstructure and tensile properties were studied. The following conclusions can be drawn:The content and morphology of α phase in microstructure are affected by BA temperature. With the increase in BA temperature the content and size of equiaxed α phase decrease and finally disappear, while the content and size of lamellar α phase increase. Thus, the structure type changes from an equiaxed structure to a lamellar structure. Therefore, reasonable BA temperature can be formulated to obtain the corresponding microstructure according to the different needs of engineering applications;It is verified that after the BASC process, only α phase and β phase are contained in the microstructure of the alloy, and no α′ phase and α″ phase is precipitated. For alloy treated by the BASCA process, the metastable β phase is decomposed;It is found that the strength of the alloy is higher after BASC process, and the microstructure is more stable after the BASCA process. By comparison, it is found that the comprehensive properties of the alloy are best when the BA temperature is 940 °C. After BASC process treatment, *R*_m_, *R*_p0.2_, *A*, and *Z* can achieve 1083 MPa, 950 MPa, 14.5%, and 28%, respectively, and these properties can achieve 1054 MPa, 874 MPa, 13.5%, and 35%, respectively, after the BASCA process treatment;By analyzing the fracture morphology it is found that the fracture morphology of the equiaxed structure is mainly dimples. The fracture morphology of the lamellar structure is mainly crystalline, and there are a small number of shallow dimples with obvious dissociation steps and tearing edges.

## Figures and Tables

**Figure 1 materials-15-08249-f001:**
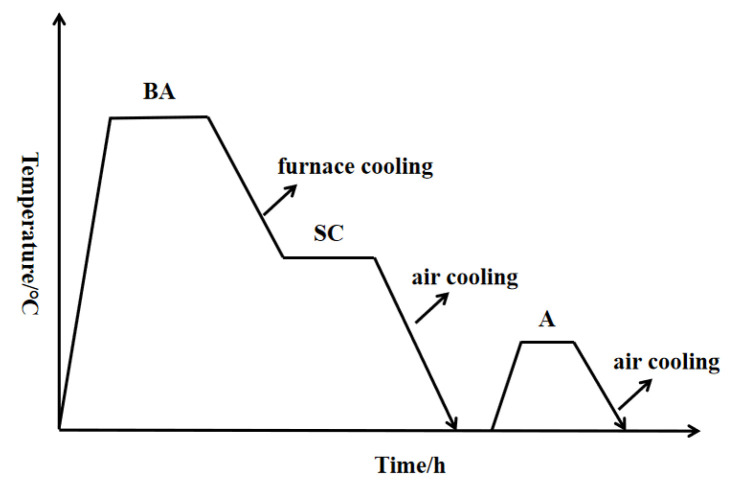
BASC and BASCA heat treatment diagram.

**Figure 2 materials-15-08249-f002:**
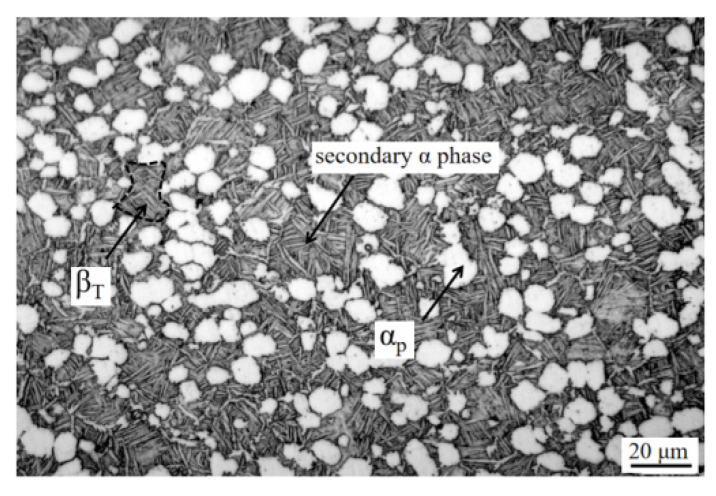
Original microstructure of TC10 alloy bars.

**Figure 3 materials-15-08249-f003:**
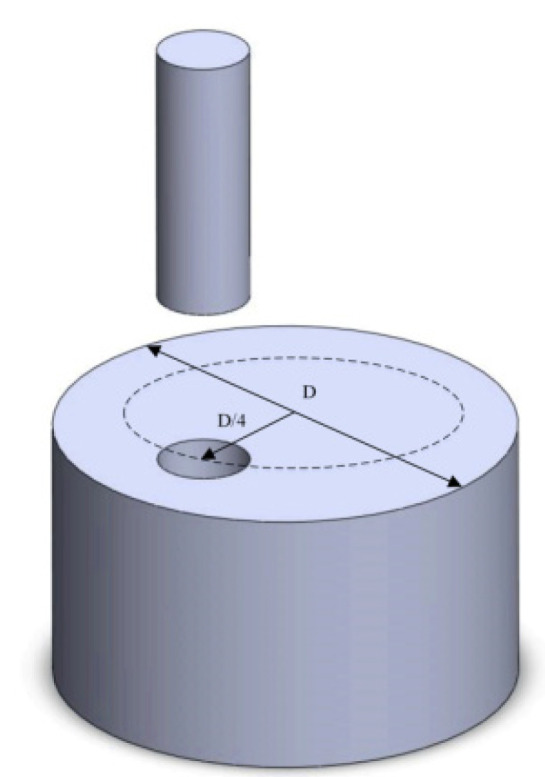
Sampling position of tensile specimen.

**Figure 4 materials-15-08249-f004:**
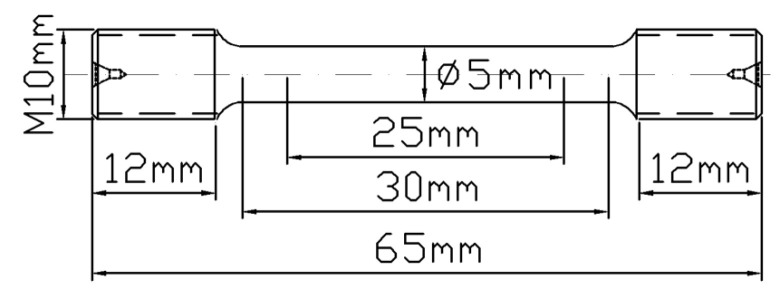
Geometry of the tensile specimen.

**Figure 5 materials-15-08249-f005:**
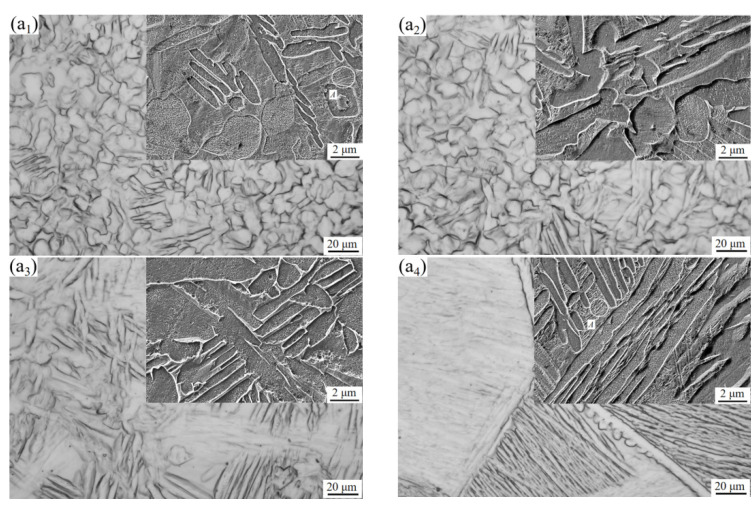
Microstructure after BASC heat treatment: (**a_1_**) 900 °C; (**a_2_**) 920 °C; (**a_3_**) 940 °C; (**a_4_**) 960 °C.

**Figure 6 materials-15-08249-f006:**
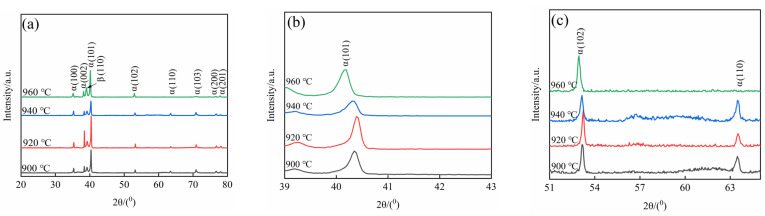
XRD diffraction patterns after BASC heat treatment: (**a**) Scanning angle (20~80°); (**b**) Scanning angle (39~43°); (**c**) Scanning angle (51~65°).

**Figure 7 materials-15-08249-f007:**
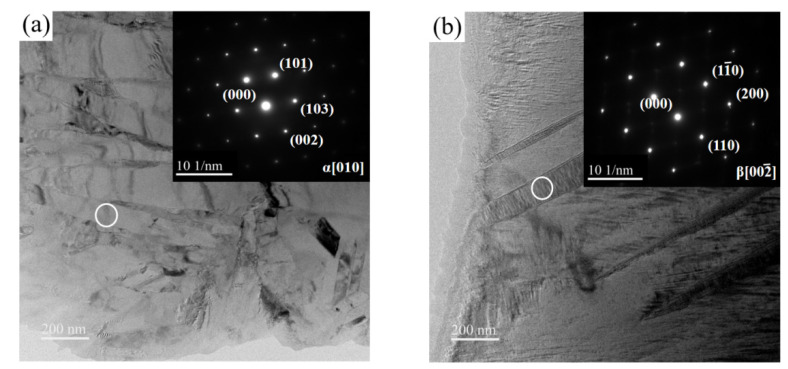
TEM morphology and selected area electron diffraction pattern: (**a**) α phase; (**b**) β phase.

**Figure 8 materials-15-08249-f008:**
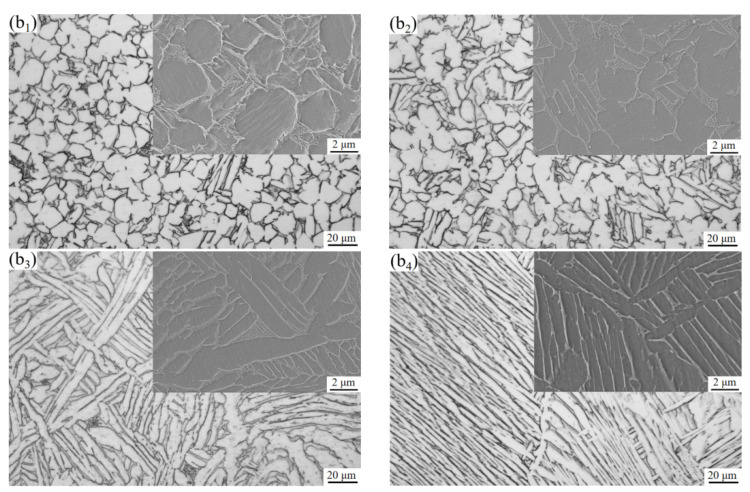
Microstructure after BASCA heat treatment: (**b_1_**) 900 °C; (**b_2_**) 920 °C; (**b_3_**) 940 °C; (**b_4_**) 960 °C.

**Figure 9 materials-15-08249-f009:**
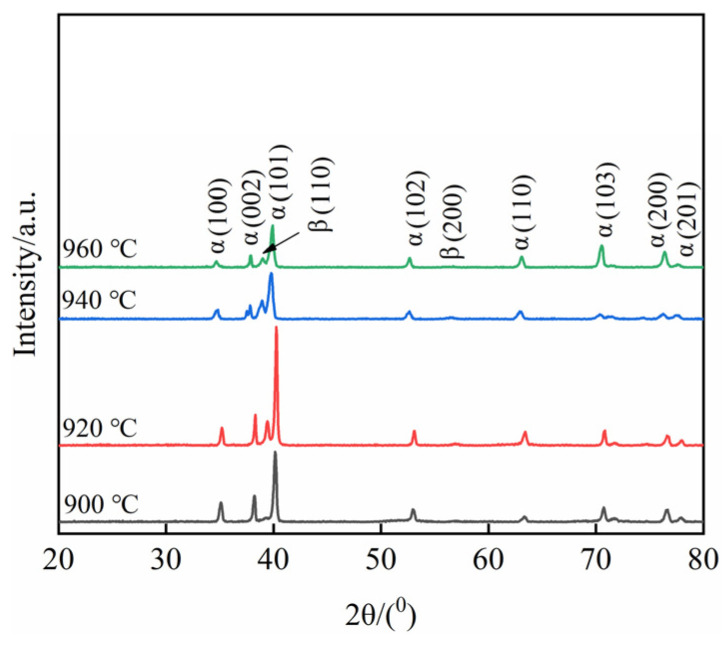
XRD diffraction patterns after BASCA heat treatment.

**Figure 10 materials-15-08249-f010:**
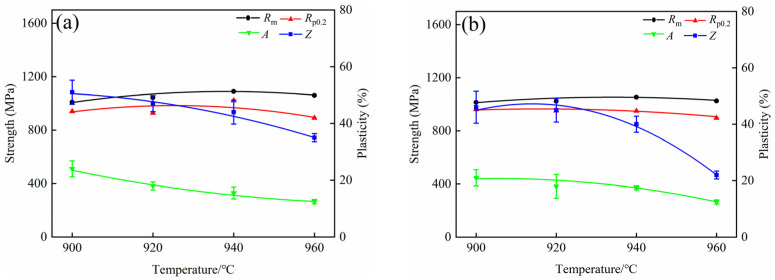
Tensile properties after BASC and BASCA heat treatment: (**a**) BASC heat treatment; (**b**) BASCA heat treatment.

**Figure 11 materials-15-08249-f011:**
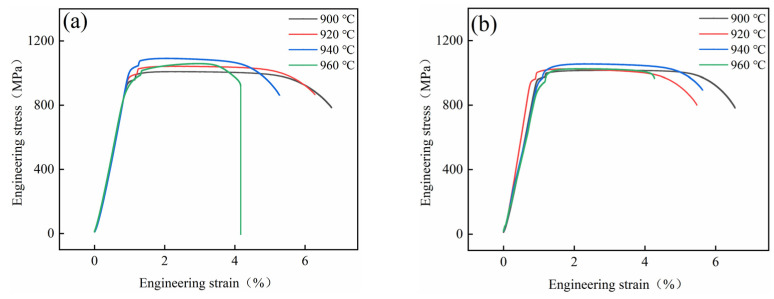
Engineering stress–strain curves after BASC and BASCA heat treatment: (**a**) BASC heat treatment; (**b**) BASCA heat treatment.

**Figure 12 materials-15-08249-f012:**
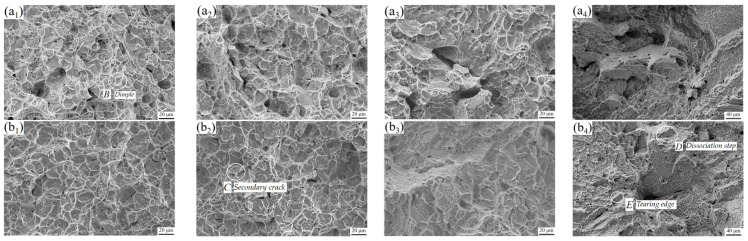
Micromorphology of tensile fracture after BASC and BASCA treatment BASC heat treatment: (**a_1_**–**a_4_**); BASCA heat treatment: (**b_1_**–**b_4_**).

**Table 1 materials-15-08249-t001:** BASC and BASCA heat treatment system.

Process	Sample	Heat Treatment System
BASC	a_1_	900 °C × 1.5 h/FC → 800 °C × 1.5 h/AC
	a_2_	920 °C × 1.5 h/FC → 800 °C × 1.5 h/AC
	a_3_	940 °C × 1.5 h/FC → 800 °C × 1.5 h/AC
	a_4_	960 °C × 1.5 h/FC → 800 °C × 1.5 h/AC
BASCA	b_1_	900 °C × 1.5 h/FC → 800 °C × 1.5 h/AC+560 °C × 4 h/AC
	b_2_	920 °C × 1.5 h/FC → 800 °C × 1.5 h/AC+560 °C × 4 h/AC
	b_3_	940 °C × 1.5 h/FC → 800 °C × 1.5 h/AC+560 °C × 4 h/AC
	b_4_	960 °C × 1.5 h/FC → 800 °C × 1.5 h/AC+560 °C × 4 h/AC

## Data Availability

Not applicable.

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
