# Peer review of "Effect of BASC and BASCA Heat Treatment on Microstructure and Mechanical Properties of TC10 Titanium Alloy"

_materials, 2022, doi:10.3390/ma15228249_

Round 1
Reviewer 1 Report
The article is about effect of BASC and BASCA heat treatment on microstructure and mechanical properties of TC10 titanium alloy. However, some issues must to be addressed:
- Abstract: Please start by expressing the aim of this paper, followed by the rest of the information. Also, please define or try to avoid using abbreviations in the abstract. Typically, the abstract should provide a broad overview of the entire project, summarize the results, and present the implications of the research or what it adds to its field.
- The bibliographic foundation is important and well executed, however some new discussions should be inserted, authors should consider some new works in the literature, such as: DOI 10.1016/j.jmbbm.2020.104198 or DOI 10.1016/j.matchemphys.2020.123959.
- Please avoid bulk citation, like 1-3, 4-8 …
- Lines 63-66: please provide sufficient details about obtaining the samples, including the size of them.
- Enhance the clarity of the figure 4.
- In discussion section you must to refere also to Young modulus of Ti alloys.
- The results are merely presented, not properly discussed. Please add explanations for the observed changes. Please give an extended discussion on the obtained results and correlate your findings with previous literature studies and prospective applications.
- More analysis and interpretation of the results should be added for a clearer understanding of observed experimental phenomena.
- The authors must to provide some details about importance of the research and their applicability.
- Please rewrite the conclusions in a more quantitative form and enhance the clarity of the conclusion section in order to highlight the results obtained.
- General check-up and correction of the English language is suggested. There are still some minor typos and grammatical errors.
The author needs to address the abovementioned points for the betterment of the manuscript.
Author Response
Response to Reviewer 1 Comments (The modified article is in the attachment)
1 Abstract: Please start by expressing the aim of this paper, followed by the rest of the information. Also, please define or try to avoid using abbreviations in the abstract. Typically, the abstract should provide a broad overview of the entire project, summarize the results, and present the implications of the research or what it adds to its field.
Reply: Thanks for the reviewer’s kind suggestions and the abstract has been rewritten. Abbreviations were also avoided in the abstract section. Please check the revised.
2 The bibliographic foundation is important and well executed, however some new discussions should be inserted, authors should consider some new works in the literature, such as: DOI 10.1016/j.jmbbm.2020.104198 or DOI 10.1016/j.matchemphys.2020.123959.
Reply: Thanks for the reviewer’s kind suggestion. New literature and discussions have been added in the revised manuscript (Lines 366-373).
3 Please avoid bulk citation, like 1-3, 4-8 …
Reply: It has been modified according to the opinions of reviewer.
4 Lines 63-66: please provide sufficient details about obtaining the samples, including the size of them.
Reply: The sample details have been added (Lines 67-68), and the sampling position and the size of the tensile specimen are shown in Figure 3 and Figure 4 (Lines 89-90).
5 Enhance the clarity of the figure 4.
Reply: It has been modified according to the reviewer’s suggestion (The order of the pictures in the paper has been adjusted, and now it is figure 6)
6 In discussion section you must to refere also to Young modulus of Ti alloys.
Reply: Thanks for the reviewer’s kind suggestion. Young modulus is usually used as an important indicator in the field of medical titanium alloys. However, in the engineering field, Young modulus is rarely used as a reference index, because the Young modulus of each type of titanium alloy is very close. In this paper, TC10 titanium alloy is mainly used in the engineering field, so Young modulus is not referred to in this paper. In the published articles, there are also few discussions on it [1-4].
[1] DOI 10.19476/j.ysxb.1004.0609.2018.04.05,
[2] DOI 10.13373/j.cnki.cjrm.xy17090024,
[3] DOI 10.3390/MET11040556,
[4] DOI 10.1016/j.msea.2013.12.024.
7 The results are merely presented, not properly discussed. Please add explanations for the observed changes. Please give an extended discussion on the obtained results and correlate your findings with previous literature studies and prospective applications.
Reply: Thanks for the reviewer’s kind suggestions. The discussion of tensile properties, the comparison between the present work and previous literature studies, and the prospective applications have been added in the revised manuscript (Lines 230-268 and Lines 269-276). Please check the revised.
8 More analysis and interpretation of the results should be added for a clearer understanding of observed experimental phenomena.
Reply: Thanks for the reviewer’s kind suggestion. More analysis and interpretation of the results are added in Lines 230-268.
9 The authors must to provide some details about importance of the research and their applicability.
Reply: At present, β-type titanium alloy is used as the research target of BASCA heat treatment, and there is no relevant report on the application of this process to α+β type titanium alloy. Therefore, the research results of this paper have shown significance for the expansion of BASCA heat treatment and α+ β-type titanium alloy (Lines 45-57).
As for the details, in this paper, the BASCA heat treatment process is divided into two processes: BASC and BASCA. The microstructure and mechanical properties of the alloy after different process treatment are analyzed, and the different characteristics of the alloy after two process treatment are pointed out. According to the actual engineering requirements, different process parameters can be formulated to obtain the required microstructure and properties (Lines 57-65, Lines 274-276).
10 Please rewrite the conclusions in a more quantitative form and enhance the clarity of the conclusion section in order to highlight the results obtained.
Reply: Thanks for the reviewer’s kind suggestion. The conclusion has been rewritten according to the requirements of the review expert.
11 General check-up and correction of the English language is suggested. There are still some minor typos and grammatical errors.
Reply: Thanks for the reviewer’s kind suggestion. The manuscript has been carefully polished and is now suitable for readers.
Reviewer 2 Report
In this article, the authors investigated the effects of BASC and BASCA heat treatment on the microstructure and mechanical properties of TC10 titanium alloy were studied by means of by microscopic measurements, X-ray diffraction, TEM and tensile test.
1. Fig.9(a), (b) : Explain why the kink were occurred around 1% engineering strain.
2. p.8 L244 : Is Fig. 9 not to be Fig. 10?
3. p.8 L264-266, “In general, the number and size of dimples can reflect the plasticity of alloy. When the number and size of dimples are large, the plasticity is good, when the number and size of dimples are small, the plasticity is poor.”
: Explain these physical causes.
Author Response
1. Fig.9(a), (b) : Explain why the kink were occurred around 1% engineering strain.
Reply: In the process of tensile test, the tensile rate is 0.00025 s-1 at the initial stage. After the tensile sample exceeds the yield point, the tensile rate becomes 0.0067 s-1 until the end of the test.
Both tensile rates are within the standard range and will not affect the tensile results. This phenomenon is caused by the change of tensile rate. (The order of the pictures in this paper has been adjusted, and now it is figure 11)
2. p.8 L244 : Is Fig. 9 not to be Fig. 10?
Reply: Thanks for the careful review of the review expert. (The order of the pictures in this paper has been adjusted, and now it is figure 12)
3. p.8 L264-266, “In general, the number and size of dimples can reflect the plasticity of alloy.
When the number and size of dimples are large, the plasticity is good, when the number and size of dimples are small, the plasticity is poor.”
: Explain these physical causes.
Reply: The larger the size and the more the number of dimples, indicating that the plastic deformation during the formation of dimples is more serious, the deformation is more sufficient, the fracture process will absorb more energy, and the plasticity is better.
Round 2
Reviewer 1 Report
The article is suitable for publication.